# Dietary Advanced Glycation End Products: Their Role in the Insulin Resistance of Aging

**DOI:** 10.3390/cells12131684

**Published:** 2023-06-21

**Authors:** Manuel Portero-Otin, M. Pia de la Maza, Jaime Uribarri

**Affiliations:** 1Departamento de Medicina Experimental, Facultad de Medicina, Universidad de Lleida, 25196 Lleida, Spain; manuel.portero@mex.udl.cat; 2Centro de Nutricion y Diabetes, Departamento de Medicina, Clinica Alemana, Universidad del Desarrollo, Santiago 7610658, Chile; mariapiadelamaza@gmail.com; 3Department of Medicine, Icahn School of Medicine at Mount Sinai, New York, NY 10021, USA

**Keywords:** insulin resistance, glycation, oxidative stress, inflammation, ultraprocessed foods

## Abstract

Insulin resistance (IR) is commonly observed during aging and is at the root of many of the chronic nontransmissible diseases experienced as people grow older. Many factors may play a role in causing IR, but diet is undoubtedly an important one. Whether it is total caloric intake or specific components of the diet, the factors responsible remain to be confirmed. Of the many dietary influences that may play a role in aging-related decreased insulin sensitivity, advanced glycation end products (AGEs) appear particularly important. Herein, we have reviewed in detail in vitro, animal, and human evidence linking dietary AGEs contributing to the bodily burden of AGEs with the development of IR. We conclude that numerous small clinical trials assessing the effect of dietary AGE intake in combination with strong evidence in many animal studies strongly suggest that reducing dietary AGE intake is associated with improved IR in a variety of disease conditions. Reducing AGE content of common foods by simple changes in culinary techniques is a feasible, safe, and easily applicable intervention in both health and disease. Large-scale clinical trials are still needed to provide broader evidence for the deleterious role of dietary AGEs in chronic disease.

## 1. Introduction

Insulin resistance (IR) is a pathophysiological condition in which organs—mostly skeletal muscle, adipose tissue, and liver—do not respond at an adequate rate to insulin, and it is considered to be a consequence of the disruption of different molecular pathways affected by insulin in these tissues [1]. In the general population, sensitivity to insulin-mediated glucose disposal in several tissues varies greatly [2]. The major consequence of IR, type 2 diabetes, arises when people who are insulin-resistant are unable to maintain the level of hyperinsulinemia required to correct the insulin action deficiency. Clinically, it appears as a defect in insulin-mediated glucose control in tissues, prominently in the above named muscle, fat and liver. Primary characteristics of IR are inhibited lipolysis in adipose tissue, impaired glucose uptake by muscle and inhibited gluconeogenesis in liver [3]. Therefore, IR also encompasses defects in lipid metabolism, in line with the multifaceted roles of insulin in metabolism regulation [4]. IR is one of the earliest manifestations of a constellation of human pathologic conditions that include metabolic syndrome, type 2 diabetes, cardiovascular diseases and aging [5]. Lifestyle modifications, including reduced intake of ultraprocessed foods containing advanced glycation (AGEs) and lipo-oxidation end products (ALEs), body weight loss and increased physical activity, have been shown to increase insulin sensitivity, thereby preventing IR [6,7].

Whether total caloric intake with body fat accretion or the presence of specific nutrients or diet-derived insulin-signaling disruptors is mostly responsible for the IR of aging is unclear [8]. Of the many dietary factors that may play a role in the aging-related lack of or decreased insulin sensitivity, AGEs appear potentially important. Recent clinical data suggest that food-derived AGEs may contribute to IR [9]. In this review, we discuss our current understanding of the association between dietary AGEs and IR. First, we describe generalities about dietary AGEs and how they could influence IR, followed by a detailed discussion of in vitro, animal and human evidence linking dietary AGEs with the development of IR. We then review some of the controversies surrounding the assessment of IR in human studies and finalize with concluding remarks.

## 2. What Advanced Glycation End Products (AGEs) and Their Biological Actions Are, and How They Can Produce Insulin Resistance

AGEs are a heterogeneous group of compounds resulting from different pathways, the classical one being the Maillard reaction, in which the carbonyl group of a reducing sugar reacts spontaneously (nonenzymatically) with the free amino group of proteins. AGEs form continuously in the body under physiologic conditions, but their rate of formation is markedly enhanced in the presence of high glucose and increased oxidative stress [10]. Of importance, AGEs are also formed in foods, and their formation is highly dependent on the temperature during processing and/or cooking of food.

One of the problems in this field is that there are many identified AGEs, but no general agreement on which one is biologically the most important. For example, very recent data show that AGE-induced inactivation of insulin-receptor substrate 1 and decreased phosphorylation of AKT, instrumental in leading to IR, are strongly dependent on the AGE structure, with high molecular weight exerting a higher pathogenic effect than small AGEs [11]. Furthermore, there are several methods available to measure AGEs, from the very simple and nonspecific autofluorescence methods to ELISA methods utilizing a variety of commercially available antibodies [12]. The most specific methods use mass spectrometry, but unfortunately are more costly, require an elaborated laboratory setup, and are more difficult to set up to measure large number of samples.

Western diets are very rich in AGEs as a result of the application of heat during cooking and the widespread processing of food. High-heat application under dry cooking conditions, such as grilling, leads to increased formation of AGEs, while cooking with lower temperatures and high water content, such as stewing, poaching and boiling, decreases the formation of AGEs. In other words, a sample of food would have very different AGE content just in response to different cooking methods [13]. About 10–30% of food AGEs are absorbed from the gastrointestinal tract into the systemic circulation, mainly as amino acids or small peptides containing AGEs. Recent data, however, show that AGE products formed in situ in the digestive system can account for a significant modification of proteins [14]. Of importance, the biological properties of AGEs may depend on the digestion step, with digested AGEs showing more proinflammatory properties than undigested AGEs [15]. Most of the absorbed AGEs are normally eliminated by the kidneys into the urine. Besides urinary excretion, there are other mechanisms regulating tissue and circulating levels of AGEs, such as AGE receptor 1 (AGER1), which catabolizes AGEs, and glyoxalases, enzymes that break down AGE precursors [16]. Most food AGEs escape digestion and absorption and end up in the colon, where they affect local microbiota metabolism and gut integrity [17]. The local colonic action of unabsorbed food AGEs may be as important in the overall proinflammatory actions in the body as those compounds directly absorbed into the systemic circulation [18,19,20,21,22]. The recent demonstration of endocytosis of dietary AGEs by enterocytes [23] suggests the existence of unmet needs in research evaluating the mechanisms linking dietary AGEs and pathogenic effects. In fact, an emerging area of nutrition research is the role of diet and bacterial metabolites in regulating gut homeostasis and inflammation, and thereby, indirectly, IR [24]. Diet-related metabolites engage metabolite-sensing G-protein-coupled receptors, including GPR41 and GPR43, which are expressed in a variety of cell types, including gastrointestinal, adipose, and immune cells. These metabolites include short-chain fatty acids produced by the microbial fermentation of dietary fibers in the colon [25]. Functions of GPR41 and GPR43 include the regulation of energy intake and expenditure, modulation of glucose metabolism, and the resolution of inflammatory responses via, for example, activation of the NLRP3 inflammasome [26,27]. By inhibiting the growth of short-chain fatty acid-producing bacterial species in the colon [28,29], dietary AGEs may contribute to heightened inflammatory signals in the gastrointestinal tract and other tissues, thereby increasing chances for IR to develop. AGEs have also been found to selectively increase in vitro expression of histone deacetylases known to be upregulated in the pathogenesis of diabetes complications [30]. This area of research, however, is still in its infancy, and further studies must explore whether dietary AGEs have the capacity to negatively regulate metabolite sensors in the gut.

There are no structural or functional differences between exogenous and endogenous AGEs. Once inside our internal milieu, they all seem to induce biological effects by the same two general mechanisms: one through direct alteration of protein structure and therefore function (and cross-linking between different proteins) and the other indirectly by binding to receptors. There is a large variety of receptors, including Toll-like receptors, scavenger receptors, G-protein-coupled receptors, and pattern recognition receptors, that interact with specific AGE structures, leading to modulation of several intracellular processes [31,32]. The receptor for advanced glycation end products (RAGE) is one of these and is the most studied protein interacting with AGEs. It belongs to the immunoglobulin superfamily, which was discovered and given its name because of its capacity to bind to AGEs [33]. However, more than a classical receptor, it may be better described as a pattern recognition receptor [34]. Binding of AGEs to RAGE initiates a series of intracellular pathways that eventually lead to the generation of reactive oxygen species (ROS) and several inflammatory transcription factors and cytokines, all of which contribute to the tissue damage associated with AGEs [10]. Specifically, persistent hyperglycemia, renal insufficiency and probably dietary AGEs raise AGE burden in the body. These compounds, when interacting with RAGE, cause a variety of signaling events with potential impact on insulin signaling: activation of mitogen-activated protein kinase (MAPK), p38, stress-activated protein kinase/c-Jun N-terminal kinase (SAPK/JNK), Ras-mediated extracellular signal-regulated kinase (ERK1/2), and Janus kinase signal transducer and activator of transcription (JAK/STAT), with subsequent activation of transcription factors such as NF-κB, STAT3, HIF-1α, and AP-1 [35]. Briefly, inducing IR by negatively regulating insulin signal transmission, JNK activation enhances the phosphorylation of the insulin receptor substrate (IRS-1) at serine residues [36]. Relevant downstream insulin signaling events, such as the phosphatidylinositol 3-kinase/protein kinase B (PI3K/Akt) pathway’s enzymatic activity, are decreased because of the phosphorylation of serine residues in the insulin receptor and IRS-1 molecule. Additionally, NF-kB is released when inhibitor of NF-kB (IkkB) proteins are phosphorylated and degraded by the proteasome, an event mediated by the RAGE-transduction-initiated IkkB kinase pathway. When NF-kB is activated, it can translocate to the nucleus and increase the expression of several inflammatory cytokines (IL-1, IL-6, and TNF-α), which can lead to IR [36]. In addition to maintaining chronic low-grade inflammation, ongoing NF-kB activation also positively controls RAGE expression by interacting with its proximal promoter region [37], closing a pathogenic circle. We know that RAGE also activates the NLRP3 inflammasome, a key player of the innate immune system [38,39]. Thus, in addition to the interleukin secretion indicated above, NLRP3-induced caspase-1 cleavage impacts the secretion of inflammatory cytokines IL-1β and IL-18 [38]. NLRP3 expression is implicated in IR in humans [40]. Moreover, RAGE enhances de novo synthesis of NF-κB p65, further fueling levels of transcriptionally active NF-κB [41]. This NF-κB subunit, through its inhibitory binding to the Slc2a4 gene promoter that codifies GLUT4 protein, could contribute to IR by diminishing the levels of this glucose transporter in skeletal muscles [42]. A relationship between dietary AGEs and innate system activation is further supported by demonstrating that complement component C1q subcomponent subunit A may be a novel AGE-binding protein in human serum [43]. In preclinical models, thermally processed diets have been shown to cause tissue inflammation and injury via the powerful proinflammatory effector molecule complement 5a (C5a), which also activates the innate immunological complement system. In fact, in a rat model of diabetes, a diet high in resistant starch fiber preserved the integrity of the intestinal barrier through suppression of complement [44]. These findings illustrate the pathways through which processed foods trigger chronic disease-causing inflammation and illustrate the complexity of diet–inflammation interactions [45], which could include local enteral changes at the microbiota and intestinal permeability levels and also immune modulation of the innate system.

In summary, enteral AGES, orally absorbed AGEs, or AGE precursors may contribute to the pathogenesis of IR by different underlying mechanisms: (1) direct modification of signaling molecules, such as insulin itself, which reduces its biological activity and affinity for the insulin receptor and therefore impairs insulin signaling [46], or the modification of the three arginine residues in the AMP binding site of AMP kinases, decreasing their activity [47]; and (2) interference with activation of downstream proteins involved in cell insulin signaling, including IRS 1 and Akt [48], through RAGE-dependent induction of proinflammatory cytokines and reactive oxygen species (ROS) [49].

## 3. Evidence of an Association between Dietary AGEs and IR

### 3.1. Animal Data

A role of dietary AGEs as a causative agent in IR has been well documented in several studies in different mouse strains by independent teams. We review some of these studies here and in Table 1. Reduced AGE intake leads to lower levels of circulating AGEs and to improved insulin sensitivity in the db/db mouse IR model [50]. To demonstrate this, db/db mice were randomly placed for 20 weeks (more than 50% of their usual life span) on a diet with either low AGE content (LAGE) or a 3.4-fold higher content of AGEs (HAGE). LAGE mice showed lower fasting plasma insulin levels and body weight compared with HAGE mice, despite equal caloric intake. LAGE mice had improved responses to both glucose (at 40 min, *p* = 0.003) and insulin (at 60 min, *p* = 0.007) tolerance tests, which correlated with a doubling of glucose uptake by adipose tissue. LAGE mice had twofold lower serum carboxymethyllysine (CML) and methylglyoxal (MG) concentrations and a better-preserved structure of pancreatic islets compared with HAGE mice [50]. Thus, the effect of dietary AGEs affects multiple tissues (liver, adipose tissue, pancreas), leading overall to impaired metabolism.

To overcome a potential effect of the db/db genotype itself, another group focused on euglycemic mice. Nontransgenic C57/BL6 mice were randomly assigned to high-fat diets (35% g fat) to induce IR, with either high (HAGE-HF group; 995.4 units/mg AGE) or low (by 2.4-fold LAGE-HF group; 329.6 units/mg AGE) dietary AGE content for 6 months (approximately 20% of the usual life span) [51]. At the end of 6 months, 75% of the HAGE-HF mice had become diabetic, while none of the LAGE-HF mice had, despite a similar rise in body weight and plasma lipids. Moreover, the HAGE-HF group showed markedly impaired glucose and insulin responses during glucose tolerance tests and euglycemic and hyperglycemic clamps and abnormal pancreatic islet structure and function compared with those of LAGE-HF mice. These findings demonstrate that the development of IR and type 2 diabetes during prolonged high-fat feeding in mice are linked to the excess AGEs/ALEs in fatty diets [51].

In addition to type 2 diabetes, IR has been associated with cognitive dysfunctions, such as Alzheimer’s disease [54]. To determine whether dietary AGEs promote aging-related cognitive decline, mice were subjected to different levels of AGEs in their diets for 18 months (i.e., 60% of their life span) [52]. Those mice in the high-AGE diet developed metabolic syndrome (with IR), increased brain amyloid-β42, intracerebral deposits of AGEs, gliosis, and cognitive deficits, accompanied by suppressed expressions of SIRT1, nicotinamide phosphoribosyltransferase, AGE receptor 1, and PPARγ. These changes were not due to aging or caloric intake, as none of them were present in age-matched, pair-fed low-AGE mice. The animal data were strengthened by the demonstration of significant temporal correlations between high circulating AGEs, impaired cognition, and insulin sensitivity in elderly subjects [55]. Clinically, it has been postulated that IR could explain part of the age-related decrease in cognitive function [56,57].

Considering the importance of the microbiota in IR [58,59] (see above), yet another study showed that a high-AGE diet induced IR and altered the gut microbiota composition and structure, reducing its diversity in mice [28]. The authors postulated that the loss of butyrate-producing bacteria in the AGE-loaded animals might have impaired the colonic epithelial barrier, thereby triggering chronic low-grade inflammation and possibly IR [28].

An involvement of AGEs in general, not necessarily dietary AGEs, in the development of IR mediated by alteration of sphingolipid metabolism was demonstrated in two different models of IR in mice, one genetically diabetic and the other diet-induced IR (fed a 60% trans-fat diet). Supplementation of a group of mice with pyridoxamine that lowered AGE levels reduced the development of IR [53].

All in all, these independent studies highlight the potential role of dietary AGEs as modifiable agents in the development of IR in mice, acting on several factors (peripheral insulin function, microbiota, cellular stress responses).

### 3.2. Epidemiological Evidence Linking Dietary AGEs and IR in Humans

In a large cross-sectional study conducted in young healthy Slovakian individuals of both sexes (*n* = 2769) IR, assessed through the Quantitative Insulin Sensitivity Check Index (QUICKI), was associated with serum and urinary levels of some α-dicarbonyls (AGE precursors, such as methylglyoxal) and AGEs, independently of cardiometabolic risk markers and sex [60]. In an American cohort, an increased association of very high dietary AGE intake (defined as the top quartile) and metabolic syndrome was described in adolescents aged 12–19 years from NHANES (years 2003–2004 and 2005–2006) [61]. The latter study also demonstrated that very high dietary AGE intake was significantly associated with three of five criteria for metabolic syndrome: waist circumference, serum triglyceride, and HDL cholesterol levels [61].

A meta-analysis of 17 randomized controlled trials comprising 560 participants also demonstrated that IR, measured by HOMA-IR, was significantly reduced in a low-AGE compared to a high-AGE diet, although there was no significant difference in fasting insulin, 2 h insulin and insulin area under the curve results between both diets after an oral glucose tolerance test [62]. This apparent contradiction stresses the need for adequate standardization of the methods to define IR, as is discussed in detail later in this chapter [63].

### 3.3. Randomized Controlled Interventional Studies Testing the Association between Dietary AGEs and IR in Humans

Table 2 describes eight independent clinical trials [64,65,66,67,68,69,70,71] that have looked at the effects of an AGE-restricted diet on IR markers, comprising a total of more than 440 participants. Six of the studies demonstrated an association between decreased dietary AGE intake and improved insulin sensitivity. One study, in healthy subjects, indicated that a low-AGE diet led to decreased serum levels of AGEs in parallel with HOMA-IR [64]. Five other studies looked at the effect of a low-AGE diet on IR in overweight and/or metabolic syndrome subjects. In four of these studies, the low-AGE diet was associated with lower serum or urinary levels of AGEs, as well as parallel decreases in IR as assessed by HOMA-IR in three of them [65,67,68] or improved glucose uptake assessed by euglycemic clamp in the fourth study [66].

One of the studies was a randomized 6-week prospective intervention in type 2 diabetes subjects with a standard diet (*n* = 13) versus low-AGE diet (*n* = 13), which showed a significant decrease in TNF-α and malondialdehyde levels in the low-AGE diet group, but without a significant change in HOMA-IR or serum AGEs [71]. Since circulating AGE levels appear to be useful surrogates of dietary AGE exposure, their lack of change during this study raises the possibility that the dietary intervention might not have been effective.

A second study listed in Table 2 also failed to show a relationship between dietary AGE restriction and changes in insulin sensitivity [69]. However, several key facts, mainly related to differences in the composition and methods to measure AGEs of the experimental diets, could help clarify the inconclusive outcome of this study in comparison with other studies cited in Table 2. For example, mass-spectrometry-based AGE measurement may differ profoundly in the content of several bio-accessible AGE products assessed by immunological measurements such as ELISA. Moreover, in those studies showing a clear effect of dietary AGEs in IR, dietary AGEs were delivered mostly by ingestion of meats and animal products, highly processed and cooked at high temperatures, while in this negative study [69], most dietary AGEs came predominantly from cereals and were based on dietary frequency in a specific Dutch population [72]. Processing of samples before AGE measurement, particularly the inclusion or not of delipidation, may also be relevant, as AGE content in dietary lipids is important [73]. All these points emphasize that standardization is a key factor in designing dietary interventions. Of interest, individual factors, such as ethnicity and age, may also be relevant in interpreting insulin sensitivity tests [74,75]. Genotypic characteristics have been shown to influence interaction with dietary AGEs (for example, FADS2 (definer) polymorphisms) [76]. Lastly, intervention length is key in interpretating data employing methods with a high interindividual variation, such as the hyperglycemic–euglycemic clamp, which in the case of the cited negative study showed a 50% standard deviation over the mean values [69]. In fact, a prior randomized controlled trial employing isotope-based euglycemic clamps showed that insulin sensitivity changed only after a six-month intensive weight-loss and exercise program [6].

### 3.4. Clinical Trials with Mediterranean and Vegan Dietary Patterns Support the Association between Dietary AGEs and IR in Humans

The above studies (Table 2) show that—generally, although not universally—reducing dietary AGE intake can diminish IR markers. One of the limitations of these studies is that they are of short duration. A longer study, CORDIOPREV, indirectly evaluated the potential impact of dietary AGEs in IR [77]. Reduction in serum AGE levels in subjects following a Mediterranean diet as part of the CORDIOPREV study, lasting for 5 years, was shown to increase significantly the probability of type 2 diabetes remission, a hard measure of IR. All participants had previous cardiovascular events and type 2 diabetes when recruited [77].

In a cohort of overweight subjects (*n* = 244) randomly assigned to an intervention with a low-fat plant-based diet (*n* = 122) or a control diet (*n* = 122) for 16 weeks, dietary AGE consumption decreased on a low-fat plant-based diet, and this was associated with changes in body weight, body composition, and insulin sensitivity, independently of energy intake [78].

Therefore, the epidemiological data and most of the interventional studies in humans support a long-term reduction in dietary AGEs being associated with an improvement in clinically relevant outcomes (insulin sensitivity, weight loss, probability of type 2 diabetes remission) requiring minimal time for revealing its beneficial effects.

## 4. Controversies about Documenting IR in Human Studies

One of the controversies about human studies dealing with the effect of dietary AGEs on IR cited above is that most of them have quantified IR using the HOMA-IR index. Hyperinsulinemic–euglycemic glucose clamp is the gold standard for directly determining insulin sensitivity, but it requires a constant insulin infusion until reaching higher steady-state insulin levels that enhance glucose disposal in skeletal muscle and adipose tissue and inhibits hepatic glucose production [4]. Euglycemia is “clamped” through infusion of 20% dextrose. However, the procedure is time-consuming, labor-intensive, expensive and technically demanding [79]. Apart from several multiple-sample methods, such as the oral and intravenous glucose tolerance tests and the Matsuda index, two simpler methods are more widely employed, requiring only fasting blood glucose and insulin determinations. One of them is the HOMA-IR, which estimates IR by a simple mathematical equation using only fasting blood glucose and insulin [80]. Similarly, the quantitative insulin sensitivity check index (QUICKI) is a simple variation of the HOMA equation showing a better correlation with the glucose clamp and other methods [81,82]. These indices, however, have limitations, and they may not accurately reflect the complex interplay of physiological and biochemical factors involved in IR. For example, HOMA-IR does not account for postprandial insulin secretion, and thus may not be suitable for evaluating IR in individuals with insulin secretion disorders or those with advanced liver or kidney disease. Furthermore, it does not identify whether hepatic or peripheral insulin resistance predominates or whether the cutoff value globally employed (2.5) is appropriate for the population under study, also considering the pulsatile nature of insulin secretion [83]. Despite these limitations, the HOMA index remains a useful tool for evaluating IR and has frequently been used in many well-controlled epidemiological studies unrelated to dietary AGEs [84].

## 5. Final Conclusions

In recent years, numerous small clinical trials have measured the effects of a low-AGE dietary intervention on a variety of clinical conditions. These trials suggest that a simple low-AGE dietary intervention, in a sustained manner, decreases circulating levels of AGEs, markers of inflammation and oxidative stress in healthy, chronic kidney disease, and diabetic patients, and improves IR, defined mostly by HOMA-IR, in diabetic and prediabetes patients. These human data in combination with strong evidence in many animal studies have generated a new paradigm of disease widely unrecognized, suggesting that excessive consumption of dietary AGEs secondary to a “Western lifestyle” represents an independent risk factor for inappropriate chronic mild oxidative stress and inflammation during life, which over time facilitates the emergence of the chronic diseases of modern world, especially diabetes and CVD. Simultaneously, we are seeing that an increasing number of people are consuming modern processed foods laden with AGEs [18]. Reducing the AGE content in common foods by simple changes in culinary techniques is a feasible, safe, and easily applicable intervention in both health and disease. Large-scale clinical trials must be performed to replicate the small clinical trials that have been performed so far and provide broader evidence for the deleterious role of dietary AGEs in chronic disease. If this evidence continues to be demonstrated, then reduction of AGE content in common foods may convey an enormous public health impact: an opportunity for a safe, inexpensive, and effective dietary modulation to prevent or improve diabetes and its secondary comorbidities.

Furthermore, this detailed review of the data suggests the existence of unmet needs in research evaluating the mechanisms linking dietary AGEs and pathogenic effects. Therefore, there is ample room for further refinement of interventional studies aimed at evaluating the effects of diets with a high AGE content in human health: the use of more robust methods for evaluation of insulin signaling (including isotope-based clamp and functional imaging) and a thorough standardization of dietary AGE components, including not only total AGE contents but also low-molecular-weight dicarbonyl precursors, which have relevant, yet unknown, roles in human pathogenesis.

## Figures and Tables

**Table 1 cells-12-01684-t001:** Selected animal studies showing an association between dietary AGEs and IR.

Author, Reference	Animal Model	Study Design	Intervention	Findings
Hofmann [50]	Db/Db mice (5 week old)	Dietary intervention with random assignment into two parallel groups for 20 weeks (*n* = 20)	High versus low AGE diets	Lower body weight, lower serum AGEs, better response to both glucose and insulin tolerance tests and better preservation of pancreatic islets than with the high AGE diet
Sandu [51]	C57/BL6 female mice (6 week old)	Dietary intervention with random assignment into two parallel groups for 6 months	High fat (35% fat) high AGE diet (HAGE-HF) versus High fat low AGE diet (LAGE-HF)	None of the LAGE-HF mice became diabetic, while 75% of HAGE-HF did.
Cai [52]	C57/BL6	Dietary intervention with random assignment into three parallel groups for 18 months	Pair-fed three diets throughout life: (1) low AGE (MG^−^) *, (2) MG supplemented low AGE chow (MG^+^) and regular chow (Reg)	Older MG^+^ and Reg fed mice developed IR (higher fasting insulin levels and abnormal intraperitoneally glucose tolerance test) and dementia, which did not happen in MG^−^ mice.
Wang [28]	C57/BL6 male mice (12 week old)	Dietary intervention with random assignment into three parallel groups for 24 weeks	Three parallel diets: (1) regular chow (*n* = 10), (2) regular chow + MG (*n* = 15) or (3) heat-treated chow (*n* = 15)	IR (high fasting insulin, HOMA and abnormal intraperitoneal glucose tolerance test) developed in groups 2 and 3, but not 1. Microbiota was also altered in groups 2 and 3 (not in group 1) leading to loss of butyrate-producing bacteria
Mastrocola [53]	Db/Db and C57/BL6 mice	Dietary intervention followed by pharmacological intervention in the C57/BL6 mice	C57/BL6 mice were randomly assigned to 4 groups for 12 weeks: (1) standard diet, (2) high fat diet (60%trans-fat), (3) standard diet + pyridoxamine for last 8 weeks, (4) high fat diet + pyridoxamine for last 8 weeks	High levels of AGEs and RAGE and abnormal enzymes of sphingolipid metabolism were found in the liver of Db/Db and group 2 C57/BL6, but not in groups 1, 3 and C57/BL6

MG * = methylglyoxal.

**Table 2 cells-12-01684-t002:** Summary of selected clinical trials evaluating the effect of an AGE-restricted diet on insulin resistance.

Author, Year, Reference	Study Design	Intervention	Number of Participants	Randomized	Participant Characteristics	Duration and Allocation	Specified Outcomes	Findings
Birlouez, 2010 [64]	Crossover	High- and low-AGE diets	62	Yes	Healthy individuals	1 month, France	Changes in serum AGEs and HOMA	Decreased serum AGEs and HOMA
Uribarri, 2011 [70]	Parallel	High- and low-AGE diets	18	Yes	Patients with diabetes	3 months, USA	Changes in serum AGEs, markers of OS and inflammation and HOMA	Decreased levels of serum AGEs, markers of OS *, inflammation and HOMA
Luevano-Contreras 2013 [71]	Parallel	High- and low-AGE diets	26	Yes	Patients with diabetes	1.5 months, Mexico	Changes in serum AGEs, markers of OS and inflammation and HOMA	Decreased markers of OS and inflammation, but no changes in serum AGEs or HOMA
Mark, 2014 [65]	Parallel	High- and low-AGE diets	74	Yes	Overweight women	1 month, Denmark	Changes in urinary AGEs and HOMA	Decreased urinary AGEs and HOMA
De courten 2016 [66]	Crossover	High- and low-AGE diets	20	Yes	Overweight individuals	0.5 months, Denmark	Changes in serum AGEs and insulin resistance (hyperinsulinemic–euglycemic clamp and intravenous glucose tolerance test)	Decreased serum AGEs and insulin resistance
Vlassara, 2016 [67]	Parallel	High- and low-AGE diets	138	Yes	Patients with metabolic syndrome	12 months, USA	Changes in serum AGEs, markers of OS and inflammation and HOMA	Decreased levels of serum AGEs, markers of OS, inflammation and HOMA
Goudarzi 2020 [68]	Parallel	High- and low-AGE diets	40	Yes	Patients with metabolic syndrome	2 months, Iran	Changes in serum AGEs, markers of OS and inflammation and HOMA	Decreased serum AGEs, markers of OS, inflammation, HOMA and weight (were also calorie restricted)
Linkens 2022 [69]	Parallel	High- and low-AGE diets	82	Yes	Patients with obesity	1.5 months, Netherlands	Changes in serum AGEs, markers of OS and inflammation and insulin resistance (hyperinsulinemic–euglycemic and hyperglycemic clamp)	Decreased circulating AGEs but not markers of OS/inflammation or insulin sensitivity

OS * Oxidative stress.

## Data Availability

Not applicable.

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
