# Peer review of "Dietary Advanced Glycation End Products: Their Role in the Insulin Resistance of Aging"

_cells, 2023, doi:10.3390/cells12131684_

Round 1

Reviewer 1 Report

This narrative review  from Portero-Otin and colleagues aim to review the current literature on dietary advanced glycation endproducts (AGEs) and the development of insulin resistance (IR). AGEs seem to play an important role in the progression of metabolic disorders and development of concurrent co-morbidities. However, the specific role of AGEs in the development of IR is not clearly established and the role of dietary AGEs compared to endogenous formed AGEs remains unclear. 

While I find the topic of the review both relevant and important, I have several concerns regarding the manuscript.  

1.    The structure of the manuscript needs major revision.

·        In general, a clearer structure separating introduction to the pathophysiology from the findings (in vitro, in vivo, RCT and epidemiological), discussion and conclusion is needed. E.g., in the introduction three different mechanisms are suggested for the AGE dependent development of IR. However, these are not revisited in the following sections and new mechanisms are introduced throughout the manuscript.

·        For the introduction I further suggest a clear description of dietary AGEs considering the proportion of dietary AGEs entering the circulation pool of compounds and the contribution to the total AGE burden.

·        In continuation of the above, a distinction between dietary AGEs per se and pathophysiological mechanisms related to AGEs in general is warranted.

·        Section 2.4. starting on line 146 includes a discussion of the methods used to measure AGEs. Although of great importance to the field it should not be included in the section reporting results on Mediterranean and vegan diets.

2.    Section 3 starting on line 165 needs major revision as the description of IR and the methods used for assessment of IR is unclear and, in some instances, misleading or wrong. Furthermore, this section lacks references.

3.    Finally, I acknowledge the unsystematic format of the review, however,  it would strength the credibility of the work if the authors indicate the conducted literature search and the applied eligibility criteria. 

4.    For Table 1 I recommend separating the columns into study design, intervention, specified outcomes and findings to get a clearer and more objective overview. 

The language needs revision for correct wording and more concise and clear formulations. E.g., line 183-184 and line 234-237 are examples of sentenses that need clearification.

Furthermore, consider when findings indicate associations and when causation is establisehd. 

Reviewer 2 Report

Comments to the authors:

The motive of the present study is interesting, and the results may be essential for us to know more about AGEs, IR, and aging. The work could be a significant contribution to the field.

However, the aim of the study should be more clearly stated, and the review has to be more justified.

It would be necessary to summarize and discuss current knowledge of AGEs and high-AGE diets. The authors need to show such information in the present review.

I recommended adding the description of IR and IR in aging.

I recommend that the descriptions of human and animal studies be shown in separate tables. It will be easier to find all information on clinical /animal trials also in tables.

The descriptions could have been more transparent throughout the review, making it easy to understand.

Major comments

Page 1, the first paragraph. The aim of the study should be added.

Insulin resistance should be described more. And in addition, IR with aging.

Next, AGEs should be described more.

Page 2-3. I recommended that the description of animal and human trials should be divided in more clear way, including tables.

Page 5, the third paragraph. The descriptions of IR controversies should be less explained. There is no topic for this review. It could be moved to the limitation of the studies.

Page 6, chapter 4. I recommended that the conclusions chapter be rewritten, and the conclusion of this review should be more precise with some more practical recommendations.

Reviewer 3 Report

In this review the Authors analyse the effect of dietary AGEs on Insulin Resistance. The topic is interesting and has recently gained relevance in the scientific community. Similar review has been published in 2015 (Ottum MS, Mistry AM. Advanced glycation end-products: modifiable environmental factors profoundly mediate insulin resistance. J Clin Biochem Nutr. 2015 57(1):1-12. doi: 10.3164/jcbn.15-3) and should be cited. In this respect, this review is an update and could be suitable for publication after minor revision. 

The introduction section could be improved in the AGEs description and in their pathological effect. Some references are not relevant (see ref 1 and 3). Line 48 is missing of reference. Animal model section: line 98 reference 14 is wrong. 

Round 2

Reviewer 1 Report

After major revision I still have concerns regarding the structure of the manuscript. I appreciate the section on dietary AGEs. However, in accordance with my initial comments a structured presentation of mechanisms, findings and discussions is still missing.

Furthermore, a clear focus on the mechanisms linking AGEs and insulin resistance is warranted. The included discussion on insulin resistance and possible association with e.g., Alzheimer’s disease and microbiota seems a bit coincidental and out of context.

I still find the description of insulin resistance vague a bit misleading.

The Table 2 (previous Table 1) has improved; however, descriptions of study design and interventions are still mixed up.

Lastly the references need revision one by one as they are not appropriate in the current form.

na

Author Response

Responses to Reviewer #1 June 1st, 2023

  1. After major revision I still have concerns regarding the structure of the manuscript. I appreciate the section on dietary AGEs. However, in accordance with my initial comments a structured presentation of mechanisms, findings and discussions is still missing. 
  2. We thank this reviewers for all the suggestions that have helps us to improve this review.
  3. Furthermore, a clear focus on the mechanisms linking AGEs and insulin resistance is warranted. The included discussion on insulin resistance and possible association with e.g., Alzheimer’s disease and microbiota seems a bit coincidental and out of context.
  4. We thank the reviewer for these suggestions. We agree that the mechanisms linking dietary AGE intake, body AGE burden and insulin resistance are yet to be clearly defined, as demonstrated by the increasing number of publications in the literature devoted to this particular issue. We have amended the manuscript in order to offer more information on potential mechanisms linking increased dietary AGE intake and insulin resistance. Of note, we should consider not only the increased amount of circulating AGEs following increased intake of dietary AGE (which has been unequivocally demonstrated in all published controlled studies), but also the local enteral effects of these compounds, which may result from the interaction of dietary AGEs with enteral cells, as well as with microbiota and microbiota-derived products. Given the importance of innate immunity in this interaction, we have described even further the findings reporting insulin resistance, inflammation and dietary AGE intake in the updated manuscript. As a result, the manuscript is now significantly increased in the number of words. All the new data added/modified now is highlighted in light blue in contrast to the previous changes highlighted in yellow.
  5. Q. I still find the description of insulin resistance vague a bit misleading. 

A: We have tried to improve the definition of insulin resistance in the updated manuscript by using references from the literature. This section at the beginning in the Introduction now reads as follows:

Insulin resistance (IR) is a pathophysiological condition in which organs –mostly skeletal muscle, adipose tissue and liver- do not respond at an adequate rate to insulin, considered to be a consequence of the disruption of different molecular pathways impinged by insulin in these tissues (Li et al., Signal Transduct Target Ther 2022; 7:216 and Yeni-Komshian et al., Diabetes Care 2000; 23:171-75). In the general population, sensitivity to insulin-mediated glucose disposal in several tissues varies greatly (Yeni-Komshian et al., Diabetes Care 2000; 23:171-752. Its major consequence, type 2 diabetes, arises when people who are insulin-resistant are unable to maintain the level of hyperinsulinemia required to correct the insulin action deficiency. Clinically, it is apparent for a defect in insulin-mediated glucose control in tissues, prominently in the above named muscle, fat and liver. The primary characteristics of IR are inhibited lipolysis in adipose tissue, impaired glucose uptake by muscle and inhibited gluconeogenesis in liver (Wilcox G, Biochem Rev 2005; 26:19-39, 2005)

  1. The Table 2 (previous Table 1) has improved; however, descriptions of study design and interventions are still mixed up.
  2. We have now updated Table 2
  3. Lastly the references need revision one by one as they are not appropriate in the current form.
  4. All references have now been updated/reviewed

Reviewer 2 Report

 Accept in present form

Author Response

No response necessary

Reviewer 3 Report

In the revised version of the manuscript the introduction section is improved. Nevertheless, the Authors forgot to update the references section and it is not possible to check the changes.

Moderate editing is required

Author Response

Response to Reviewer #3 June 1st 2023

  1. In the revised version of the manuscript the introduction section is improved.Nevertheless, the Authors forgot to update the references section and it is not possible tocheck the changes.
  2. all the references have now been updated/reviewed

Q: Furthermore, a clear focus on the mechanisms linking AGEs and insulin resistance is warranted. The included discussion on insulin resistance and possible association with e.g., Alzheimer’s disease and microbiota seems a bit coincidental and out of context.

A: The updated manuscript has attempted to further define these potential mechanisms. All the new data added/modified now is highlighted in light blue in contrast to the previous changes highlighted in yellow.

Round 3

Reviewer 1 Report

I find the manuscript much improved and the structure clearer. I appreciate the section on mechanisms but for the benefit of readability and uniformity with the rest of the manuscript, I suggest shortening it and point to the overall regulatory mechanisms without detailed descriptions of all the signaling pathways. 

Please consider shorter sentences and more concise descriptions. 

Author Response

Comments and suggestions for Authors

I find the manuscript much improved and the structure clearer. I appreciate the section on mechanisms but for the benefit of readability and uniformity with the rest of the manuscript, I suggest shortening it and point to the overall regulatory mechanisms without detailed descriptions of all the signaling pathways. 

We thank the reviewer for his comments.

We have slightly shortened the section on mechanisms taking into consideration that further reduction would have reduced the mandatory manuscript length of more than 4,000 words.

Comments on the Quality of English Language

Please consider shorter sentences and more concise descriptions. 

We have tried to use shorter sentences throughout the manuscript.